

# Is retroflexion a stable cue for distributional learning for speech sounds across languages? Learning for some bilingual adults, but not generalisable to a wider population in a well powered pre-registered study

Hannah L. Goh[1,2], Luca Onnis[3,4] and Suzy J. Styles[2,5]

[1] Interdisciplinary Graduate Programme, Nanyang Technological University, Singapore, Singapore
[2] Psychology, School of Social Sciences, Nanyang Technological University, Singapore, Singapore
[3] Center for Multilingualism in Society across the Lifespan, University of Oslo, Oslo, Norway
[4] Department of Linguistics and Scandinavian Studies, University of Oslo, Oslo, Norway
[5] The Centre for Research and Development in Learning (CRADLE), Nanyang Technological University, Singapore, Singapore

Corresponding author
Hannah L. Goh,
hannahle001@ntu.edu.sg

## ABSTRACT

Bilinguals are widely reported to have certain kinds of cognitive advantages, including language learning advantages. One possible pathway is a language-specific transfer effect, whereby sensitivity to structural regularities in *known languages* can be brought to *to-be-acquired languages* that share particular features. Here we tested for transfer of a specific linguistic property, sensitivity to retroflexion as contrastive phonemic feature. We designed a task for bilinguals with homogeneous language exposure (*i.e.*, bilingual in the same languages) and heterogeneous feature representation (*i.e.*, differing levels of proficiency). Hindi and Mandarin Chinese both have retroflexion in phoneme contrasts (Hindi: stop consonants, Mandarin: sibilants). In a preregistered study, we conducted a statistical learning task for the Hindi dental-retroflex stop contrast with a group of early parallel English-Mandarin bilinguals, who varied in their Mandarin understanding levels. We based the target sample size on power analysis of a pilot study with a Bayesian stop-rule after minimum threshold. Contrary to the pilot study ($N = 15$), the main study ($N = 50$) did not find evidence for a learning effect, nor language-experience variance within the group. This finding suggests that statistical effects for the feature in question may be more fragile than commonly assumed, and may be evident in only a small subsample of the general population (as in our pilot). These stimuli have previously shown learning effects in children, so an additional possibility is that neural commitment to adults' languages prevents learning of the fine-grained stimulus contrast in question for this adult population.

## INTRODUCTION

Bilinguals tend to show a learning advantage for acquiring novel language structures. For example, bi- and multilinguals show enhanced learning for grammatical structures (*Kemp, 2007*; *Kovács & Mehler, 2009*), acquiring new vocabulary (*Kaushanskaya & Marian, 2009*; *Yoshida et al., 2011*), and developing sensitivity to phoneme contrasts (*Antoniou et al., 2015*; *Singh et al., 2018*; *Wang & Saffran, 2014*). In addition, bilinguals may be better at learning novel language-like structures in artificial grammar learning tasks (*Onnis, Chun & Lou-Magnuson, 2018*). While the exact mechanisms underlying this advantage continue to be explored, we have identified three main overarching hypotheses in existing literature. We refer to these hypotheses as domain-general advantages, general linguistic advantages, and specific linguistic advantages.

Firstly, *domain-general* accounts propose that acquiring multiple languages may lead to the development of cognitive enhancements that boost learning for novel structures–including linguistic structures. Support for a *general bilingual advantage* comes from cognitive studies comparing groups with different language backgrounds. For instance, studies have found that bilinguals may have significantly better executive control and selective attention than monolinguals (*e.g.*, *Bialystok & Craik, 2010*; *Carlson & Meltzoff, 2008*). However, it is unclear whether such domain-general cognitive advantages would contribute to the specific case of language learning. Some studies have investigated correlations between executive control and language-related learning. *Yoshida et al. (2011)* reported a bilingual advantage and correlations between the performance of 3-year-olds on language and executive function tasks. However, *Bartolotti et al. (2011)* found mixed evidence, with bilingual experience influencing learning of two conflicting artificial languages only in low interference conditions, and inhibitory control influencing learning in high interference conditions. Hence, relationships between bilingualism and cognitive functions may not necessarily be linear; instead, bilingual learning advantages may merely be mediated by general cognitive functions. Furthermore, the question of whether bilingualism actually affords this kind of general cognitive advantage remains hotly debated. A recent review by *Antoniou (2019)* found that studies often disagree whether bilinguals show improved cognitive functions, and whether some groups of bilinguals demonstrate enhanced cognitive functions (*e.g.*, children and the elderly) while others (*e.g.*, young adults) do not. In current literature, therefore, it is difficult to establish how much a domain general bilingual advantage contributes to the learning of new language structures.

Secondly, *general linguistic* accounts focussing on the linguistic nature of the learning advantage propose that bilingual learning advantages are grounded in the linguistic experience of managing and making sense of multiple languages. By extension, acquiring new linguistic patterns and structures would therefore be easier for individuals who already have more than one linguistic pattern in their language repertoire. For instance, *Tremblay & Sabourin (2012)* trained adult participants on a novel Hindi dental-retroflex contrast and found that bilinguals and multilinguals showed larger learning effects and better transfer effects to a novel but related phoneme contrast than monolinguals.
Further support for the idea that linguistic structural overlap might mediate bilingual learning advantages can be seen in a study by *Enomoto (1994)* who found general bilingual advantages in sensitivity to a Japanese geminate contrast were stronger, especially when specific phonetic features overlapped with the bilinguals' linguistic experiences.

Therefore, while there is some evidence to suggest there is a *general linguistic advantage* for bilinguals, it may be the case that learning advantages are *specific to properties of the languages in question*, and may rely on the degree of overlap between the structure of the languages in a speaker's repertoire and the to-be-acquired linguistic content. For instance, a three-year longitudinal study by *Kopečková (2016)* found that individuals who were familiar with the alveolar/r/ phoneme in at least one of their existing languages (*e.g.*, German-Croatian, and German-Russian bilinguals) were better at learning to produce the Spanish alveolar trill/r/, as compared to bilinguals without similar linguistic overlaps. Similarly, *Antoniou et al. (2015)* tested English monolinguals, Mandarin–English bilinguals, and Korean–English bilinguals on their ability to learn a novel place-of-articulation contrast based on the phonology of Mandarin, and an artificial lenition contrast based on the phonology of Korean. They found that while both bilingual groups were better than English monolinguals at learning the Mandarin-based place of articulation contrast, only the Korean-speaking bilinguals showed a learning advantage for the more difficult Korean-based lenition contrast. This finding supports the hypothesis that learning advantages may be specific to linguistic overlaps between existing linguistic features and to-be-learned stimulus contrasts. There is evidence to support that fact that *language-specific bilingual advantages* may be long-lasting and can be present even if exposure to a language only occurred for a short period early in life. For instance, *Werker (1986)* demonstrated that adults who had only been exposed to Hindi up to the age of two were significantly better at discriminating a difficult Hindi dental-retroflex contrast as compared to monolingual and bilingual adult participants who did not have any experience with Hindi across the lifetime. Similar forgotten-language results were also found by *Singh & Seet (2019)*, who found that adults who had been raised by Hokkien speaking caregivers up to the age of three were significantly better at learning Hokkien tonal contrasts than adults who had only been raised by English speaking caregivers. Likewise, *Oh et al. (2010)* found that Korean–Americans adopted from Korea in early childhood were significantly better at learning Korean aspirated and lenition phonemes compared to individuals with no early childhood Korean exposure. The impact of language-specific learning effects may also extend to the way in which novel grammar is learned. *Onnis & Thiessen (2013)* exposed Korean and English adults to a statistical learning task, in which forward and backward probabilities between adjacent pseudowords generated two equally probable and orthogonal patterns in their order of presentation, one more similar to Korean, and the other more similar to English. Their results showed that participants' probabilistic preferences for pseudoword order aligned with the word order pattern that was more similar to their L1 language, indicating that their learning was heavily influenced by their prior experience of linguistic patterns. Moreover, while the Korean participants in the study had received extensive formal training in English and lived in an English-speaking environment, they nevertheless exhibited statistical learning biases

congruent with their native Korean. These findings appear to show that the strength of *specific bilingual linguistic advantages* may be predicted by the strength of representation that individuals have with each of the languages in their repertoire. On the whole, these studies demonstrate that shared phonetic and structural features appear to be powerful predictors of linguistic learning advantages, thus supporting a *specific bilingual linguistic advantage* based on familiarity of shared features or regularities.

To explore the language specific overlap hypothesis, we decided to examine whether a language specific learning effect could be found in an auditory statistical learning task for English–Mandarin bilingual adults, if the to-be-learned contrast included a phonetic feature that has contrastive functions in Mandarin. For auditory targets, we selected a Hindi dental-retroflex phoneme contrast identical to that of studies carried out by *Golestani, Paus & Zatorre (2002)*, *Golestani & Zatorre (2009)*, and *Vandermosten et al. (2018)* on children and adults. The phonetic feature of retroflexion is shared between Hindi and Mandarin Chinese, albeit in different phonemes (Hindi: stops; Mandarin: sibilants), thereby providing linguistic overlap with the phonology of one of our bilingual participants' languages. Hence, while English-Mandarin bilinguals will have familiarity with an alveolar-retroflex contrast in sibilants such as /s/ *vs.* /ʂ/, they will be unfamiliar with the Hindi dental-retroflex stop contrast of /d̪/ *vs.* /ɖ/. Furthermore, our participants–Singapore's English-Mandarin bilinguals–vary in their Mandarin proficiency, exposure, and use. As a result, we expect that there will be individual differences in Singapore's English–Mandarin bilinguals' familiarity with the retroflex feature of Standard Mandarin (*i.e.,* the Beijing accented variety), allowing us to explore the question of whether the strength of a potential language specific learning effect might be linked to strength of representation that individuals have with the overlapping feature in question.

We chose to base our study on a methodological adaptation of *Vandermosten et al. (2018)*'s passive listening statistical learning paradigm, adapted for adults. Statistical learning involves the process of learning by extracting regularities from sensory input in the environment, and plays a critical role in the process of early language acquisition, particularly in the learning of native-language phoneme contrast boundaries. According to *Werker, Yeung & Yoshida (2012)* the extraction of information from the relative frequencies of meaningfully contrastive speech sounds heard early in life is crucial to the formation of perceptual boundaries around unique native language phonetic categories. Examples of this can be seen in infant studies that track the development of phoneme sensitivities that shift from a more general ability to distinguish between a wide variety of phoneme contrasts, to becoming more aligned to infants' native languages over the course of several months in early infancy (*e.g.*, *Werker & Tees, 1984*).

The ability to learn new phoneme contrast categories through statistical learning continues throughout childhood, as children have the ability to track statistical information about frequencies of exposure to different sound tokens arranged on an acoustic continuum. For instance, when trained on a bimodal distribution of speech sounds (with more tokens drawn from steps closer to the ends of an acoustic continuum), children have been shown to develop a two-category perception of the continuum, behaving as though the continuum consists of two main contrastive categories of speech

sounds (*e.g.*, *Vandermosten et al., 2018*). However, studies of passive statistical learning in adults differ in whether they generate expected learning effects. *Terry, Ong & Escudero (2015)*, for example, found that adult Australian English speakers were unable to show significant learning effects following bimodal distributional training on the Dutch /ɑ/-/a:/ phoneme contrast, even with the use of stimuli that enhanced the differences in the vowel categories, and an event-related potential mismatch negativity study by *Wanrooij, Escudero & Raijmakers (2013)* found that Dutch-speaking adults could not discriminate between the English vowels /æ/ and /e/ following bimodal training on the vowel sounds. At the same time, *Hayes-Harb (2007)* also demonstrated that while adults may be able to extract some degree of novel phoneme category information from statistical learning, they appear to rely more heavily on lexical information in their learning of novel contrasts. On the other hand, stronger evidence of statistical learning in adults has been found in some studies. *Chládková & Šimáčková (2021)* found Czech speakers were able to learn a novel Czech-like durational contrast following bimodal distributional training while Greek speakers who were not familiar with this linguistic feature were not able to learn. Similar evidence of adults benefitting from statistical learning can also be seen in a series of studies that show that Dutch-learning native Spanish speakers living in the Netherlands show a significant improvement in their perception of the difficult Dutch /ɑ/-/a:/ vowel contrast following bimodal distributional training (*Escudero et al., 2011*; *Escudero & Williams, 2014*; *Wanrooij, Escudero & Raijmakers, 2013*). Critically, one key factor in the studies where adults *were* able to learn from statistical information is that they all had some degree of overlap in their language repertoire (phonology, word order) with the stimulus dimension that they were being trained on.

Hindi and Mandarin Chinese both have retroflexion in phoneme contrasts (Hindi: stop consonants, Mandarin: sibilants). Based on existing evidence, we thus predicted English–Mandarin bilinguals adults that would show a learning effect for a Hindi dental-retroflex contrast in a distributional learning paradigm similar to that of *Vandermosten et al. (2018)*. Therefore, we decided to base our exploration of the language specific effect on Singapore English–Mandarin bilingual adults by determining if their familiarity with the Mandarin retroflex contrast could be related to their statistical learning of retroflexion in Hindi stop consonants. We first conducted a pilot study, and then a full study powered on the basis of pilot results.

## Pilot study

In a pilot study, we tested if adult participants would be able to learn the Hindi dental-retroflex contrast with a distributional learning paradigm. To begin with, all participants were tested on their ability to discriminate the contrast. Following this pre-test, there was a training period in which participants were exposed to sounds from the Hindi-dental retroflex continuum, with presentation frequencies representing a bimodal frequency distribution. A post-test was used to establish whether learning had occurred. The methodology of *Vandermosten et al. (2018)* was adapted in the following ways: Firstly, we increased the number of sound tokens that participants heard during training, to maximise possible learning effects, while maintaining the proportion frequency of each

sound token such that this was consistent with the original *Vandermosten et al. (2018)* study. Secondly, rather than comparing results between learners and a control group trained on a unimodal distributional frequency, we used learners' performance accuracy in pre-training as a baseline for comparison. Lastly, we conducted an unattended training paradigm in which participants were told that they did not need to pay attention to the sounds, rather than the original attended auditory task, to reduce working memory load.

## METHODS

### Participants

Sixteen participants (10 female) were recruited from the student population of Nanyang Technological University in exchange for course credits. Since colour was a key feature of the test stimuli, one male participant was excluded due to colour blindness. The remaining 15 participants were aged between 20–29 years old (*Median* = 22), were ethnically Chinese Singaporeans, and reported being fluent in Mandarin and English, but not Hindi (see detailed Language ID profiles below). This study was approved by the IRB of the host institution Nanyang Technological University (IRB-2019-01-034), and written consent was obtained from all participants prior to the study.

### Stimuli

Visual stimuli were two cartoon aliens (purple and orange), and a "transmission device" that illustrated 'sending messages to aliens.' During the training phase, participants were also presented with cartoons that were muted (selected clips from Series 2 of Mr. Bean: The Animated Series; (*Atkinson & Senior, 2003*)). Audio stimuli were identical to and taken from *Vandermosten et al. (2018)*. As we did not create these stimuli, details on the creation of the auditory tokens can be found in *Golestani, Paus & Zatorre (2002)*. As in the original study, the stimulus continuum consisted of seven sound tokens 220 ms in duration. Sound 1 represents a dental /d̪/, and Sound 7, a retroflex /ɖ/, synthesized in the original study to be close to the category boundary for native speakers of Hindi. The original stimulus set also included intermediate audio tokens (Sounds 2–6) synthesized at evenly spaced acoustic intervals between these two tokens, making a continuum from the /d̪/ to the /ɖ/. Each consonant in the stimulus set was followed by the vowel /a/, making a dental-retroflex da continuum of seven sound tokens. All stimuli were presented on OpenSesame (*Mathôt, Schreij & Theeuwes, 2012*). Code for the study is archived in the repository for this paper (*Goh, Styles & Onnis, 2020*).

## PROCEDURE

### Familiarization phase

The goal of this phase was to introduce the participants to the two phoneme categories, and to allow them to familiarise themselves with the sound-to-alien pairing. In an onscreen task, participants were told that two aliens make different sounds. On each familiarisation trial, participants were presented with an image of one of the aliens at a random position on the screen while a sound was played. An orange alien was paired with
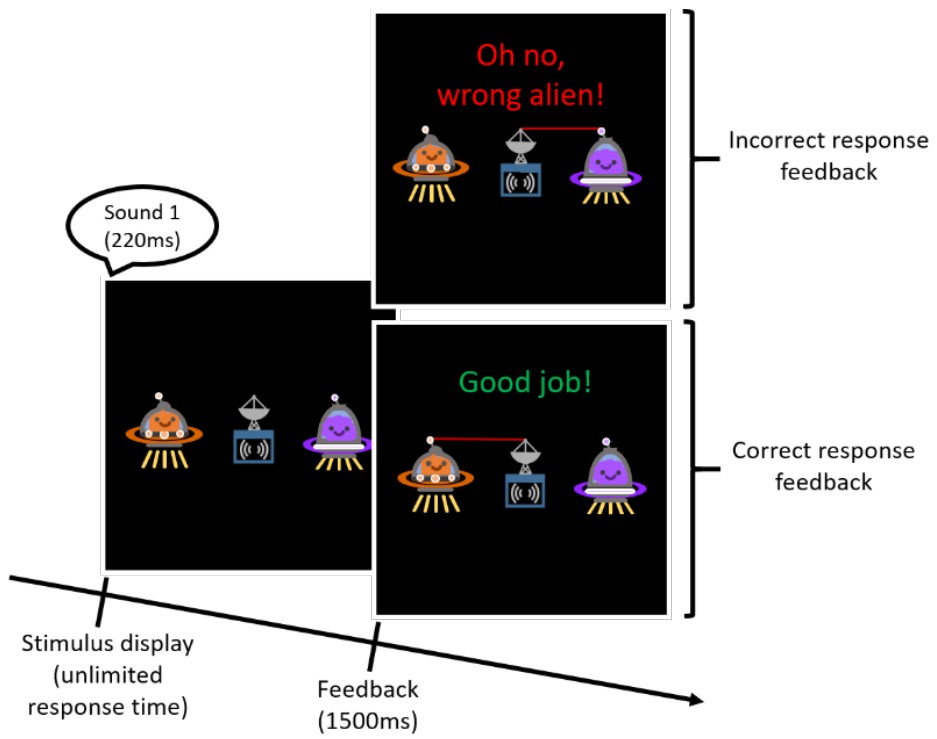

**Figure 1** Schematic of stimuli presented during a practice trial.

Sound 1 (the dental /d̪/ end of the spectrum), and a purple alien was paired with Sound 7 (the retroflex /ɖ/ end of the spectrum). Participants completed one familiarisation block in which each alien was presented 10 times in alternating order. No response was required during the familiarisation block.

## Practice phase

To ensure that participants were aware of the sound-to-alien pairing, they then completed a practice block in which they were asked to 'send messages to aliens' using the pairing they had just observed. On each trial, participants saw both aliens (See Fig. 1) and one of the previously familiarized stimulus tokens was played. Participants were asked to select an alien by pressing the key below it. Keypress responses were followed by an animated 'beam' from the 'transmission device' to the selected alien, along with a 'transmission sound' (created by Freesound.org contributor Jagadamba; downloaded from https://freesound.org/people/Jagadamba/sounds/253908/), and onscreen feedback. Each stimulus token was played four times in random order over the course of the block.

## Pre-training baseline

This phase was similar to the practice phase, with the exception that participants heard all seven of the sounds on the continuum and were not given feedback following a response made. To establish a baseline for each person, participants completed three test blocks in which the stimuli for each trial were randomly selected from the full stimulus
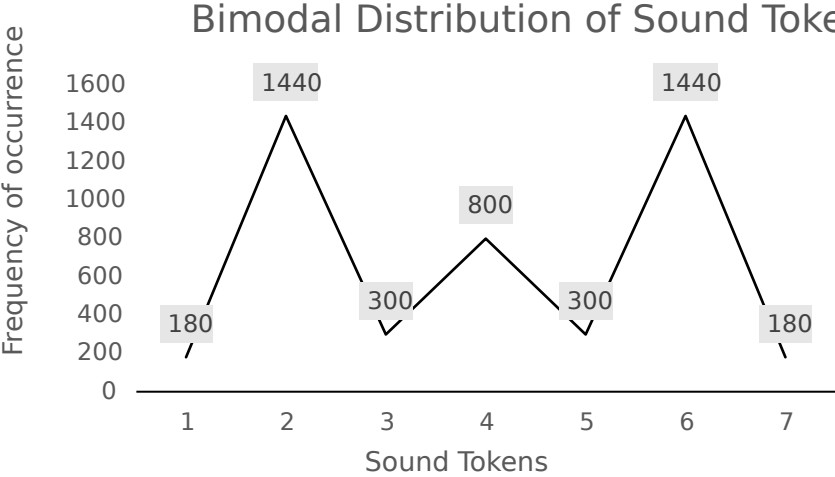

**Figure 2 Frequency of sound tokens in bimodal distribution used for training.**

continuum. Each of the seven sounds was played twice in random order over the course of each test block. There was no response time limit on each test trial. Between each test block, participants completed a mini practice block (containing only Sounds 1 and 7) to maximise their chances of remembering the correct sound/alien correspondence.

## Training phase

In the training phase, participants were asked to watch a cartoon while audio stimuli played. In this unattended paradigm, a total of 4,020 sound tokens were presented to participants in a random order, with an 80 ms ISI between each sound. To achieve a bimodal stimulus distribution, each of the seven sounds from the stimulus continuum was played according to the distribution in Fig. 2. The training phase lasted around 20 min in total.

## Post-training test

After the unattended training phase, participants completed a second round of testing, identical to the Pre-training Baseline. No breaks were given between any of the phases.

## Language ID profile

To capture variance in participants' individual language exposure, at the end of the alien message task participants were asked to rate their proficiencies in all of the languages/dialects they understand on a Likert scale from 1, "I understand a few words in this language", to 7, "I have native-level understanding of this language". The full demographic and language questions are in the repository for this paper (*Goh, Styles & Onnis, 2020*).

## Analysis plan

We measured the number of times each participant selected the purple alien (Sound 7 retroflex /ɖ/) or the orange alien (Sound 1 dental /d̪/) for each stimulus token and computed the proportion of Sound 7 responses at each step on the stimulus continuum.

Psychometric curves were fitted for each participant, and slope values were calculated using the *quickpsy* function in R (*Linares & López-Moliner, 2016*; *R Core Team, 2020*). Steeper slopes (*i.e.,* higher slope values) represent greater sensitivity to differences between bimodally distributed sounds on the dental-retroflex continuum, while flat slopes represent chance responding. In order to compare pre-training baseline to post-training test, slopes were computed separately for the pre- and post-training test blocks. As the sample size was small, a non-parametric Wilcoxon signed-rank test was carried out to test for difference in the slope values in the pre- and post-training test blocks.

In addition, in order to check whether any changes in slope value had occurred prior to the bimodal exposure, we tested whether participants showed any differences between their first and third pre-training test blocks. Individual slope values were derived separately for the three pre-training test blocks, and a non-parametric Wilcoxon signed rank test was carried out. All analyses were carried out with the statistical software R (*R Core Tream, 2020*).

## RESULTS

### Language understanding

Participants' self-reported English understanding was high (Range: 5–7, $M = 6.5$, $SD = 0.6$) and Mandarin was more variable (Range: 3–7, $M = 5.0$, $SD = 1.3$). None of the participants reported understanding any Indian languages or other languages with retroflexed phonemes. (See distribution of participants' self-rated English and Mandarin understanding ratings in Fig. 3.)

### Distributional learning effect

To check whether any learning had taken place prior to the bimodal exposure training phase, a Wilcoxon signed-rank test compared the first and third pre-training blocks. This analysis revealed no significant difference ($Z (15) = -.03$, $p = .98$; pre-1: slope $M = 0.99$, $SD = 3.80$; pre-3: slope $M = 0.60$, $SD = 2.13$).

The fitted psychometric curves for pre-training and post-training can be seen in Figs. 4A and 4B, where it is clear that many participants show a steeper curve in the post-training phase. The Wilcoxon signed-rank test revealed a significant difference in slope values between pre-training and post-training ($Z (15) = -2.61$, $p = 0.004$; pre-training slope: $M = 0.5$, $SD = 0.19$; post-training slope: $M = 0.27$, $SD = 0.33$), indicating that learning had occurred during the unattended training phase.

## DISCUSSION

The pilot study demonstrated a statistically significant learning effect in a methodological adaptation of *Vandermosten et al. (2018)*, conducted with adults. This finding suggests that the method and the stimuli are suitable for use with bilingual adults in the local context and would be able to detect distributional statistical learning of a non-native Hindi dental-retroflex contrast among Singaporean English-Mandarin bilinguals. Furthermore, since individuals in the pilot showed a range of differences between their
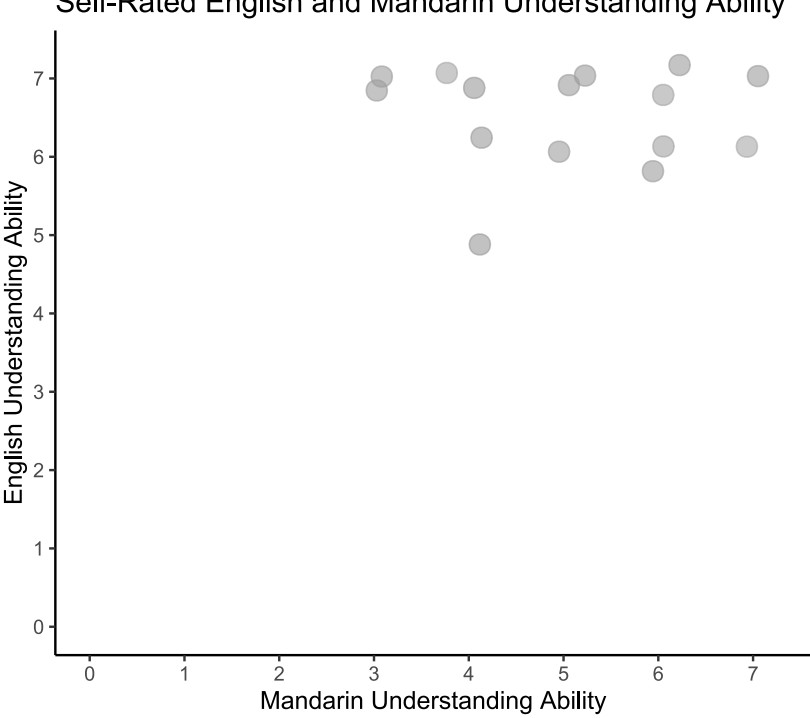

**Figure 3** **Self-rated English and Mandarin understanding ability.** Pilot participants' self-reported English and Mandarin understanding ability ($N = 15$). Responses jittered by .25 for visualization.

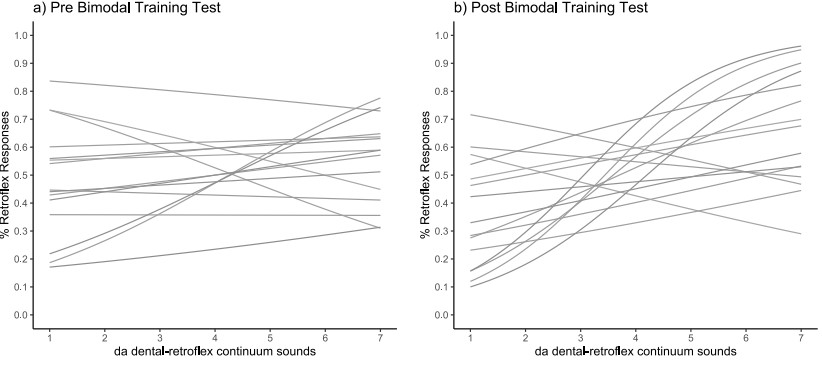

**Figure 4** **Categorisation sensitivity slopes for individual participants.** Each slope shows the fitted psychometric function in (A) pre-training test and (B) post-training test ($N = 15$).

pre- and post- training slope scores (pre-test: min $= -0.30$, max $= 0.45$; post-test: min $= -0.2$, max $= 0.82$), we observed that individuals differed in the strength of the learning effect. This means that the paradigm is suitable for further interrogation of individual differences in performance. Consequently, we preregistered a full-scale study based on these pilot findings. While our preregistration also included a Mandarin alveolar-retroflex

sibilant word identification task, we did not include it in this study. The results of the task were not included in any following analyses as there was a lack of meaningful variance in the results of the task as the majority of our participants performed at ceiling level. The results of this task have been presented in our paper on perception and production of the Mandarin alveolar-retroflex contrast in Singapore (*Goh et al., 2022*).

## MAIN STUDY

Following the pilot study, we carried out a full-scale study to further explore the language-specific hypothesis. We predicted that higher self-rated understanding of Mandarin would correspond to greater familiarity with the retroflex feature, as this feature is prevalent in the standard variety of Mandarin (*i.e.,* the Beijing variety), which is the dominant variety taught to bilinguals in the Singapore school system. If language specific exposure is a source of bilingual learning advantages, then we expected to see greater learning in individuals with higher Mandarin understanding scores.

The full-scale study was identical to the pilot, with the addition of supplementary language scales designed to identify individual differences in participants' familiarity with Mandarin. To identify a suitable sample size, we conducted an a priori power analysis in G*Power 3.1.9.7. This analysis revealed a minimum sample size of 30 would be required to observe an effect size equal to the size observed in the pilot ($dz = 0.626$) at an alpha level of 0.05 with a power of $1\text{-}\beta = 0.95$. We increased the minimum sample to $N = 50$ to ensure we had a minimum sample size suitable for an exploratory factor analysis (EFA) of language factors as recommended by *de Winter, Dodou & Wieringa (2009)*. In addition to this minimum sample size, we preregistered a data collection 'stop rule' according to a Bayesian significance test on statistical learning effects to determine if the results were substantially supportive of either H1 ($BF > 3$), or H0 ($BF < 0.33$) (*Dienes, 2014*). The prior of 0.17 was determined using the size of the measured difference between pre- and post-training slope values in the pilot study. The hypotheses and analysis plan were preregistered on the Open Science Framework (*Goh, Styles & Onnis, 2020*). The procedure was reviewed and approved by the ethics board of the host institution Nanyang Technological University (IRB-2019-01-034).

## METHODS

### Participants

In total, 56 participants (42 female) were recruited from the University student population in exchange for course credits or payment ($15) following completion of the study. Three male and three female participants were excluded from analysis, due to either a failure to follow instructions, or not being from the target demographic. The remaining sample of 50 participants were aged between 18–25 years of age ($M = 20.3$, SD $= 1.7$), all of whom were ethnically Chinese Singaporeans. Written consent was obtained from all participants prior to the study.

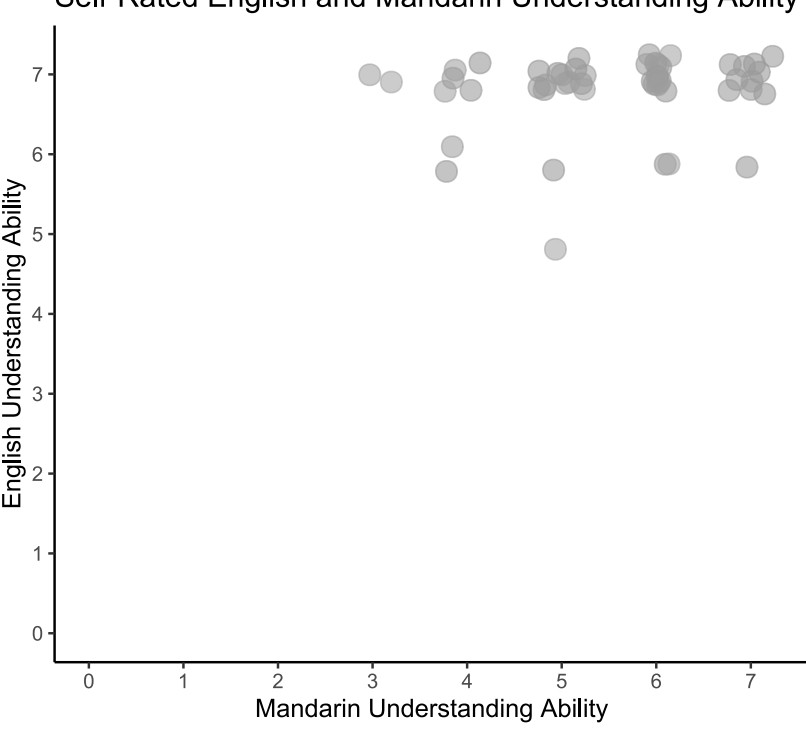

**Figure 5** **Self-rated English and Mandarin understanding ability.** Distribution of participants' English and Mandarin proficiencies in main study ($N = 50$). Responses jittered by .25 for visualization.

## Procedure

The procedure for the distributional learning and language understanding data collection portions of the study were identical to the pilot.

## RESULTS

### Language ID profiles

On average, our participants' self-reported Mandarin understanding scores were variable (Median = 6, Range: 3–7), while their English understanding was high (Median = 7, Range: 5–7), indicating that they would have had been able to understand the experiment instructions (see Fig. 5 for visualisation of English and Mandarin proficiencies in participants). Two participants also reported understanding a few words in Tamil, with both rating their Tamil understanding with a score of 1. Two variables were obtained from participants' self-rated language understanding scales for inclusion in further analyses. The Mandarin understanding score of each participant was obtained directly from the number rating that each participant gave for Mandarin understanding on the Language ID rating scale. We also checked our participants' English understanding scores to ensure that all our participants had a sufficient level of English understanding to fully understand the study instructions, all of which were presented in English.

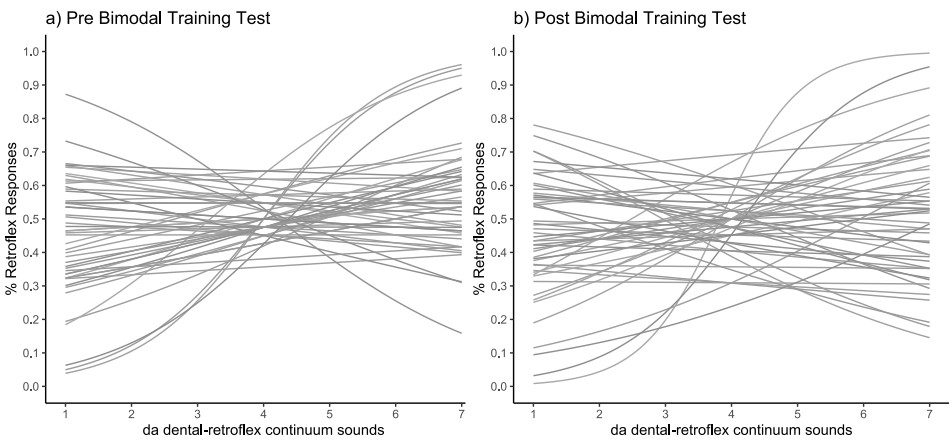

**Figure 6 Categorisation sensitivity slopes for individual participants.** Each slope shows the fitted psychometric functions for (A) pre-training test and (B) post-training test ($N = 50$).

## Distributional learning

To check whether exposure to the sound tokens heard during the pre-test blocks lead to learning before the bimodal exposure training period, we conducted a preregistered control analysis. No significant difference was observed in slope between the first and last pre-training blocks ($t(49) = 1.04$; $p = .30$; pre-1: $M = .12$, $SD = .44$; pre-3: $M = .05$, $SD = .47$), showing no significant pre-training learning. Given that no pre-training learning occurred, we proceeded to the main analysis. The fitted psychometric curves for each participant at pre- and post-training are shown in Figs. 6A and 6B. As the slope values did not fit a normal distribution, a log transformation was applied to approximate Gaussian distributions.

A multiple mixed models analysis was conducted on our participants' individual log-transformed slope values, with the categorical fixed factor of test phase (pre-training, post-training) and the interaction between test phase and self-reported Mandarin understanding rating, with random intercept of participant and random by-participant slope for test phase. Bayes factors were calculated for each of the $p$-values obtained (Bayes calculator created by *Palif (2013)*, based on the 2008 Dienes Bayes calculator, *Dienes (2008)*). Priors of 0.17 for the categorial fixed factor of test phase, and −0.29 for the interaction between test phase and Mandarin understanding rating were used for the Bayes factors with both priors being obtained from the pilot study.

Figure 6 shows the fitted psychometric functions for each individual before and after training. The analysis revealed no significant main effect of test phase (pre-training, post-training) on slope values, and the Bayes factor indicated substantial evidence in support of the null hypothesis that learning did not occur following training (pre-test: $M = .06$, $SD = .25$; post-test: $M = 0.04$, $SD = .27$; $t(1,48) = −.54$; $p = .59$, $B_{H(0,.17)} = .0001$). Figure 7 shows the derived slope values pre- and post-training, along with self-reported understanding of Mandarin Chinese. No significant interaction was observed between test phase and self-reported Mandarin understanding scores ($t(1,48) = .31$; $p = .76$;

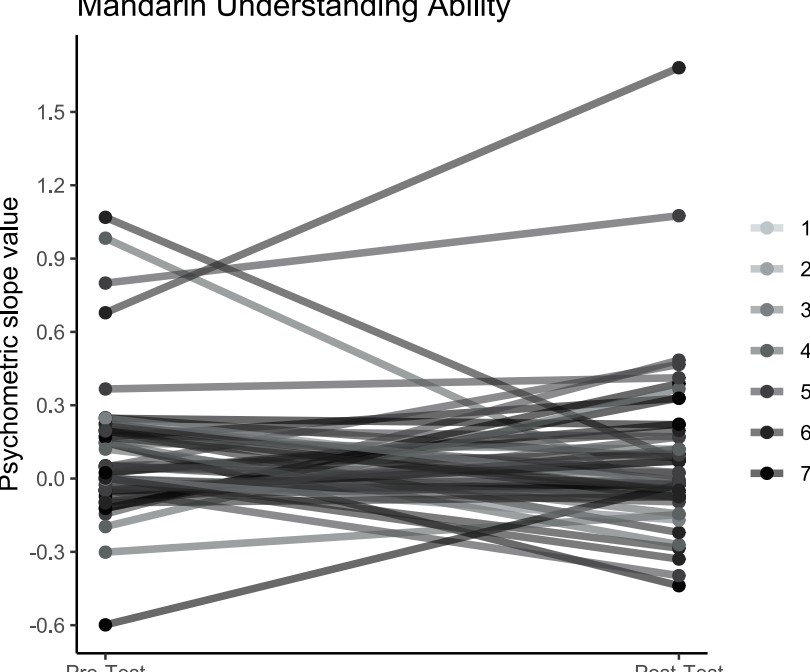

**Figure 7** **Interaction between learning effect and self-rated Mandarin understanding ability.** Fitted slope values for individuals at pre- and post- training, coloured by self-reported understanding ability of Mandarin Chinese (1 = 'a few words'; 7 = 'native').

$B_{H(0,-.29)} = .72$). Contrary to the results of the pilot study, the main study showed no evidence of learning at the group level and no evidence that individual Mandarin understanding influenced learning rate between individuals within the group, likely due to the absence of an overall learning effect.

## Preregistered exploratory factor analysis (EFA)

To find out whether language background was related in a meaningful way to the microstructure of the task—for example whether people with different kinds of language backgrounds showed learning effects over different timescales—we conducted a preregistered EFA, using the *princomp* function (*R Core Team, 2020*). In order to conduct this EFA, we derived two additional language measures from the Language ID questionnaire: the total number of languages understood by each participant, and the overall language understanding scores of each participant. We also computed three additional learning microstructure measures from the distributional learning task: 'no-training baseline', 'plasticity', and 'elasticity'. A total of six factors were entered into the EFA: Mandarin understanding score, total number of languages understood, overall language understanding score, no-training baseline, plasticity, and elasticity.

### Language measures

Total number of languages understood of each participant was computed as the total number of languages the participant rated as 1 or above on the language profile questionnaire. The overall language understanding score of each participant was computed as the sum of all the numerical ratings the participants gave for each of their language proficiencies. For example, a participant who rated their English understanding as 6 and their Mandarin understanding as 3 would have an overall language understanding score of 9. Participants reported understanding an average of 3.9 languages each (Range: 3–7, $SD = 1.3$), resulting in an average overall understanding score of 17.1 (Range: 10–31, $SD = 4.1$).

### Slope microstructure

Figure 8 shows a schematic of three possible phases in a learning trajectory, where an individual may show differences in their response pattern at a micro-structural scale, even in the absence of a global learning effect. *No-training baseline change* was designed to check if any participants exhibited any sensitivity to the target phoneme category prior to exposure training. This was obtained by deducting the slope values of the first pre-training test block from the third pre-training test block, with positive values indicating an improvement in perception prior to the exposure training phase. *Plasticity* was designed to find out whether learning was evident immediately following training, relative to the final pre-trained perceptual state. Plasticity was computed by deducting the slope values of the third pre-training test block from the first post-training test block, with positive values indicating an increase in perceptual ability immediately following the exposure training phase. *Elasticity* was designed to check for possible learning attrition over time in the post-test phase. Elasticity was computed by deducting the slope values of the first post-training test block from the third post-training test block, with a negative value indicating learning attrition. On average, participants showed a no-training baseline change of $-.07$ ($SD = .48$, range: $-1.26$–$0.75$). On average, participants showed a plasticity change of $.04$ ($SD = .60$, range: $-1.65$–$1.33$). Finally, on average, participants showed an elasticity change of $1.74$ ($SD = 14.0$, range: $-19.5$–$95.4$).

### Results

The EFA revealed only one component that accounted for more variance than one variable (*i.e.,* 20%). Component 1 revealed strong relationships between language factors. Participants who had more languages in their repertoire, and had a higher overall language understanding score were less likely to show a shift in perception prior to training. On the other hand, relationships between Mandarin understanding and plasticity were much weaker. Participants who had lower Mandarin understanding scores were moderately less likely to show a shift in perception following training—this could indicate that there was a degree of interaction between Mandarin understanding ability, and distributional learning effect for the Hindi dental-alveolar contrast. However, this EFA did not provide us with any further clarification on learning patterns of interest. See Table 1 for factor loadings on each component.

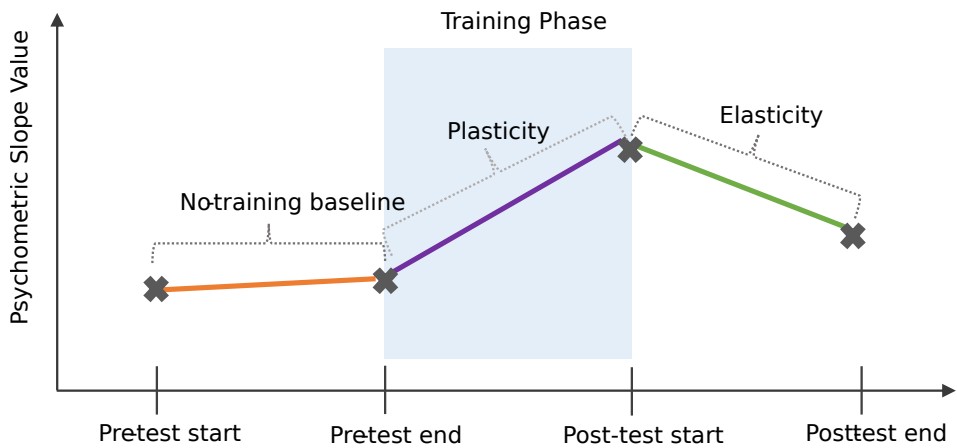

**Figure 8** Schematic of a possible learning microstructure across distributional learning task.

**Table 1** Factor loadings on exploratory factor analysis of learning microstructure and language understanding.

| | Component | | |
|---|---|---|---|
| | **1** | **2** | **3** |
| Eigenvalues | 2.41 | 1.19 | .92 |
| Percent Variance Explained | 40.2%* | 19.8% | 15.4% |
| **Factor loadings** | | | |
| No-Training Baseline | .663 | | |
| Plasticity | −.474 | | |
| Elasticity | | .687 | .507 |
| Mandarin Understanding | −.419 | .681 | −.562 |
| No. Languages Understood | −.802 | | |
| Language Understanding score | −.914 | | |

Notes.
Elasticity is inversely represented, with positive value indicating lower Elasticity.
*Indicates principal components of interest that account for >20% of variance.

## DISCUSSION

While we did observe a distributional learning effect in a small-scale pilot study, our preregistered full-scale study did not show evidence of a distributional learning effect for the Hindi dental/retroflex contrast in a well-powered study of English–Mandarin bilingual adults. One likely reason for the difference is that the learning effect in the pilot may have been driven by a small number of strong learners who are less representative of the general population. Both the pilot and main study were drawn from the student population of the host university, using the same recruitment pathway, so there is no reason to suspect different recruitment biases between the two phases of the research. Rather it seems to represent a random sampling effect in the small pilot sample. In the preregistered analysis plan, we used Bayesian hypothesis testing to qualify our results, and determine the optimal time to stop recruitment following the minimum sample

size. Unlike *p*-values which only provide an estimate of the likelihood of a false positive result, the Bayes factor estimates how much more supportive the data are of either the experimental hypothesis or the null (*Dienes, 2016*; *Lakens et al., 2018*). The Bayes factor for the main effect of phoneme contrast sensitivity (pre-training *vs.* post-training) confirmed that our data was substantially more supportive of the null hypothesis as compared to the experimental hypothesis (1,000 times more supportive of the null), allowing us to have confidence in our conclusion that learning did not occur. This difference between findings in a small pilot sample and a larger, preregistered sample is a powerful demonstration of the value of preregistration in confirmatory hypothesis testing. In addition, we demonstrate the value of Bayesian inference as a threshold for ceasing to collect data after substantial evidence against the experimental hypothesis has been accumulated.

Our pre-registered stop rule was based on the results of the main effect of learning in the pilot study, as a learning effect would first need to be obtained before we could look at the impact of Mandarin understanding on any individual differences in the strength of the observed learning. At the sample size consistent with the stop rule, the Bayes factor calculated for the interaction between Mandarin understanding and learning was not substantial. However, we conducted a supplementary linear mixed effect model on our pilot participants' individual log-transformed slope values to assess the interaction effect between self-rated Mandarin understanding and learning effect in the pilot study (fixed factor if test phase: pre *vs.* post; random intercept of participant and random by-participant slope for test phase). This analysis revealed that there was no significant interaction effect between the two factors $t(1,13) = -2.02$, $p = .06$, $\eta p^2 = .24$. A power analysis carried out on G\*Power 3.1.9.7. revealed that a minimum sample size of 44 should be sufficient to observe an interaction effect size equal to that observed in the pilot ($\eta p^2 = .24$; $f = .56$) at an alpha level of 0.05 with a power of $1 - \beta = 0.95$. Since the main study was terminated at 50 participants, the power analysis confirms that the study was well-powered to observe a significant interaction, if one had been evident.

The lack of a learning effect in the main study was somewhat surprising given the results of the pilot and previous reports of learning with the same stimuli (*Vandermosten et al., 2018*). However, as the Vandermosten study was conducted with Dutch children in Grade 3, the difference between study results may lie in the diminished ability of adults to pick up on fine-grained acoustic contrasts that children are able to detect. Indeed, some studies have shown that there may be a shift in perceptual plasticity for speech-sounds, with younger learners able to learn category structure from subtle distributional cues, and older learners showing more reliance on phonetic and lexical cues (*Hayes-Harb, 2007*; *Werker, 2018*).

In addition, prior evidence has shown that adults are poorer at detecting unfamiliar phoneme contrasts (*Best & Strange, 1992*; *Tees & Werker, 1985*). This could have meant that the dental-retroflex contrast was too difficult for our adult participants to learn within a single distributional learning paradigm alone. Some studies have suggested that learning is possible for adults if the paradigm includes exaggerated phoneme contrast hyperarticulation during training, similar to that of infant-directed speech (*Escudero*

*et al., 2011*), or in multiple sessions of bimodal distributional training (*Escudero & Williams, 2014*). However, some groups of adults are still unable to learn a completely novel phoneme contrast with bimodal training, even when the acoustic difference in categories is emphasised during training (*Terry, Ong & Escudero, 2015*). Notably, the sound tokens used in the current study were synthesised to be very finely tuned in terms of the acoustic differences between each sound, with the tokens differing only slightly from each other on the third formant and central frequency of the initial burst (*Golestani, Paus & Zatorre, 2002*). However, differences in a naturalistic Hindi dental-retroflex contrast are more complex, as *Verma & Chawla (2003)* found key differences in not only the third formant, but also in the first, second, and fourth formants in their analysis of the Hindi dental-retroflex contrast. These additional formant transition differences could play an important role in emphasising differences between the phoneme categories, leading to the synthesised continuum being harder to discriminate than typical exemplars of the naturalistic speech contrast. Hence the particular tokens used in the current study may well be beyond the sensitivity of most adults to result in learning in a passive exposure task.

Finally, while standard descriptions of both Mandarin Chinese and Hindi phonology include the retroflex place of articulation, it is possible that the tongue positions differ. In particular, Mandarin Chinese retroflexion may have less backwards curvature at the tongue tip than the retroflexion common to Hindi. Indeed, studies on Mandarin articulation have found that some Mandarin speakers use a "bunched" tongue position instead (*Ou & Guo, 2014*; *Ladefoged & Maddieson, 1996*; *Luo, 2020*). Moreover, some studies have shown that speakers of "outer-circle" varieties of Mandarin (*e.g.*, Taiwan and Singapore) tend to exhibit an alveolar-retroflex phoneme contrast merger, or deretroflexion. Indeed, one recent study documenting the alveolar-retroflex contrast in Singapore Mandarin has revealed that Singapore Mandarin speakers show signs of deretroflexion, as evidenced by smaller acoustic differences between the two categories of phonemes as compared to Beijing Mandarin speakers (*Goh et al., 2022*). Therefore, it is possible that there may be less structural overlap between Mandarin and Hindi retroflexion than would be necessary for a learning transfer effect to be observed in adults (*e.g.*, *Chládková & Šimáčková, 2021*). Taken together, this suggests that the linguistic feature canonically known as 'retroflexion' may be under-specified for the purposes of fine-grained perception tasks involving speakers of different languages. Furthermore, merely having a contrast between dental and another place-of-articulation based on tongue position (bunched or curled) is not a sufficient source of linguistic structure to enhance learning for a subtle new acoustic contrast in the general population.

While it seems a small number of individuals do show the expected effect (as in the pilot study), they were sufficiently rare in the general population that no significant interaction was observed between learning and self-reported Mandarin understanding. Indeed, some studies have also observed no relationship between Mandarin speakers' general ability to understand Mandarin and their perceptual sensitivity to the Mandarin retroflex (*Goh & Styles, 2022*), nor their production of Mandarin retroflexion (*Chung,*

*2006*; *Goh et al., 2022*), suggesting that measures of self-reported language understanding may not be sensitive to individual differences in phoneme perception.

Our small-scale pilot study was used to plan a preregistered study with sample size determined through a combination of minimum sample for statistical procedures (exploratory factor analysis) and Bayesian stop rule. The combination of these tools gives us greater confidence that although some individuals in the study did show behaviours consistent with 'learning' of the novel contrast (as in the pilot), the effect did not generalize to the broader population. This finding suggests that a language-specific advantage for learning statistical structure is not sufficiently prevalent or powerful to induce learning of a novel contrast in *all* bilingual adults when the to-be-learned contrast is acoustically subtle, and does not overlap substantially with phonetic features in their repertoire.

## CONCLUSION

Many streams of research suggest that bilinguals have certain cognitive advantages over monolinguals, and evidence is particularly compelling that learning of novel linguistic structures is enhanced by overlap between known languages and to-be-learned linguistic features. In our statistical learning paradigm, we found evidence of neither a statistical learning effect nor language specific exposure effects on learning in the general population, in our large-scale preregistered sample. However, we did observe that *some* Singaporean adult participants exhibit sensitivity to the unfamiliar Hindi contrast following bimodal training, as shown in the results of the small-scale pilot, but these individuals do not appear to be representative of the population in general. Since sensitivity to unfamiliar speech sound contrasts is known to decline with age, this paradigm likely has its maximum effect in a younger age group, such as school aged children (*Evans, Saffran & Robe-Torres, 2009*; *Vandermosten et al., 2018*), or in infants (*Maye, Werker & Gerken, 2002*; *Saffran et al., 2008*). Moreover, in order to investigate the possibility of a language-specific learning transfer effect in Singapore English-Mandarin bilinguals, training stimuli should be more closely aligned to the articulatory characteristics typical to Singapore Mandarin. Further investigation with Singaporean English–Mandarin bilingual children and a different set of training stimuli would allow us to find out whether differences in individual familiarity with phonetic features does indeed interact with ability to detect and learn linguistic contrasts.

### Funding

This research was supported by the following funding sources: Singapore's National Research Foundation under the Science of Learning (NRF2016-SOL002-011), the Centre for Research and Development in Learning, Nanyang Technological University (JHU IO 90071537). The funders had no role in study design, data collection and analysis, decision to publish, or preparation of the manuscript.

## Grant Disclosures

The following grant information was disclosed by the authors:
Singapore's National Research Foundation under the Science of Learning: NRF2016-SOL002-011.
Centre for Research and Development in Learning, Nanyang Technological University: JHU IO 90071537.

## Competing Interests

The authors declare there are no competing interests.

## Author Contributions

- Hannah L. Goh conceived and designed the experiments, performed the experiments, analyzed the data, prepared figures and/or tables, authored or reviewed drafts of the article, and approved the final draft.
- Luca Onnis conceived and designed the experiments, authored or reviewed drafts of the article, and approved the final draft.
- Suzy J. Styles conceived and designed the experiments, analyzed the data, authored or reviewed drafts of the article, and approved the final draft.

## Human Ethics

The following information was supplied relating to ethical approvals (i.e., approving body and any reference numbers):

Nanyang Technological University granted Ethical approval to carry out the study within its facilities (IRB reference number: IRB-2019-01-034).

## Data Availability

The data is available at the Open Science Framework: Goh, Hannah L, Suzy J Styles, and Luca Onnis. 2023. "Effect of Unattended Distributional Training on Phoneme Category Discrimination in English-Mandarin Bilingual Adult Participants in Singapore." OSF. April 26. doi: https://doi.org/10.17605/OSF.IO/AGRPJ.

The pilot study data is available at Zenodo: Hannah L Goh, Suzy J Styles, & Luca Onnis. (2022). Effect of unattended distributional training on phoneme category discrimination in English-Mandarin bilingual adult participants in Singapore (Pilot Study Data) [Data set]. Zenodo. https://doi.org/10.5281/zenodo.7827172. This is a mirror of the original data repository on OSF (doi: https://doi.org/10.17605/OSF.IO/AGRPJ).

The main study data is available at Zenodo: Hannah L Goh, Suzy J Styles, & Luca Onnis. (2022). Effect of unattended distributional training on phoneme category discrimination in English-Mandarin bilingual adult participants in Singapore (Main Study Data) [Data set]. Zenodo. Available at https://doi.org/10.5281/zenodo.7827176. This is a mirror of the original data repository on OSF (doi: https://doi.org/10.17605/OSF.IO/AGRPJ).

## Supplemental Information

Supplemental information for this article can be found online at http://dx.doi.org/10.7717/peerj.15467#supplemental-information.

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
