# Peer review of "Is retroflexion a stable cue for distributional learning for speech sounds across languages? Learning for some bilingual adults, but not generalisable to a wider population in a well powered pre-registered study"

_PeerJ, doi:10.7717/peerj.15467_

## Round 0.1 · original submission · Minor Revisions

The reviewer comments are quite clear and constructive, so there's no need to reiterate or belabour them. I ask that you submit a point-by-point respose to their comments to make it easier to understand how they were addressed (or why they were not).

Reviewer 1 ·

Basic reporting

First of all, this is a very well put together paper, and most of my comments would simply give the reader a smoother experience. The only major issue that stood out to me was the missing phoneme identification task mentioned in pre-registration (see comments under Experimental design).

The intro sets up 3 possible explanations for why bilinguals would be better at learning a novel language structures. The first hypothesis was clear, but the second two got a bit lost. Either listing all 3 in the initial paragraph where the explanations are set up, or consistently italicizing each of the 3 methods (currently 2 are italicized), or naming them at the beginning of each paragraph that describes them would help the reader.

The data and analysis files shared on OSF are great! It would be good to fill in the OSF wiki in a way that makes it easy for the reader to understand what is in each folder. E.g., the 001_readme text could go directly on the OSF homepage wiki.

For the future, it’s great to make an R-markdown instead of a basic r script that you knit into an html and upload with the R file so readers can go see the code working without having to run it themselves. I don’t think it’s necessary for this review, especially given how long your analysis file is, but just something to think about for the future when first setting up the analysis file.

I don't find Figure 7 or Figure 9 to be that visually useful. I would say Figure 7 could be left off, or if you would like to keep it, remove levels 1 and 2 of Mandarin knowledge from the plot since they have no participants in them, to allow the gradation of the colors between 3-7 to vary more. At the moment it's overwhelmingly red. For Figure 9, if you decide to keep it, could you differentiate the mean from the individual data points somewhat more visually? It's difficult to see anything but a flat line for the first 2 graphs. I actually think given what little is happening, Figure 9 may be better to just stay in OSF.

Experimental design

The stimuli section only describes the consonants in isolation, but on Figure 4 and 6 the stimuli are labelled as a 'da continuum'. Please specify if there was a vowel included in the stimuli and what it was.

There is one aspect of your pre-registration that does not appear in the paper--the phoneme identification accuracy measure. Please mention somewhere in the paper why the reported results do not encompass that element of the pre-registration because it was an integral part of your original second hypothesis (H2a). If the results were uninteresting and therefore not put in the paper, you still need to put the analysis and data on OSF and make a mention of it in the paper. If you plan on publishing the results separately--you need to make it clear that this paper is only reporting part of the results from your pre-registration and honestly explain why the paper deviates from the pre-registration.

Validity of the findings

Random effects of the model aren't well described. Line 293 says "with random intercept and random slope for participants" but it's not clear which effect had a random slope for participants. (I believe it is a random by-participant slope for test phase).

It wasn't clear how many variables were being entered into the EFA from the description. An explicit mention of this, followed by a list of them would help around the paragraph after line 308.

I don't think the description of the EFA results fully captures what is in the EFA table. For example, the number of languages relating to no-training baseline effects is highlighted, yet the total language score has a larger factor loading. The EFA results summary needs to be improved with the relationship between the variables in component 1 discussed somewhat more.

Additional comments

Minor edits:
line 69: no-be-learned languages --> to-be-learned languages
line 80: learnt-->learned
line 158: mention the university name here first

·

Basic reporting

In general, the manuscript is well-written, and the figures and tables are clear, despite the following issues:

Line 27: The authors mentioned there are three hypotheses, and if I understand correctly, they are the domain-general, the general linguistic advantage, and the specific bilingual linguistic advantage accounts. It will be better if the authors can mention the three hypotheses explicitly in line 27 or in other places for clarity.

Line 77, 97, 126: Extra comma after in-text citation.

Line 102: “Singapore’s English-Mandarin bilinguals” comes up so suddenly. Maybe better add “, our current participants” after that.

Line 170: Sounds 2-5 should be sounds 2-6? (where is sound 6?)

Line 183: The format of “Fig. 1” is different from other figures (e.g., “Figure 2” in line 200).

Figure 2: The last word on the y-axis cannot be seen completely.

Line 155 and 198: Please clarify what “unattended” means. Participants were told to not pay attention to the video? Or just the sound? (I think the description in the pre-registration is much clearer)

Figure 7: “a)” in the title should be deleted?

Line 311: Full stop before “in order to”.

Figure 9: Each curve represents each participant?

Line 379: Use “;” to split in-text citation.

Line 405: In-text citation “and”  “&”

Line 436: In-text citations should appear in alphabetical order. Check all other cases.

Experimental design

Introduction and discussion parts: Given that there have already been many studies related to the language specific overlap hypothesis with contradictory results conducted as reviewed by the authors (lines 57-92), the authors may need to emphasize the significance or theoretical contribution of the current study more. For example, besides the methodological adaptations stated in lines 147-156, how is the current study different from previous studies and how does it address issues that haven’t been solved? Does it help explain some contradictory findings in previous literature? Is there any special reason to adopt dental and retroflex as stimuli, and to what extent they can generalize to other sounds?

Line 158: Why 16 participants were recruited? Was this sample size based on any power analysis or effect size from previous studies?

Line 169-171: For clarity, please briefly mention how the audio tokens were synthesized. For example, which software was used, and the steps?

Figure 2: In line 120, the authors mentioned that a bimodal distribution of speech sounds should have more tokens drawn from the ends of an acoustic continuum. However, figure 2 showed that the current experiment had the least token drawn from the ends, but most tokens were drawn from the 2nd and 6th sound tokens. Please be consistent and please state how you decided the frequency of each token.

Method: Please justify why a two-alternate forced choice task instead of a continuum task, in which participants can respond along a continuum, was used, as some previous studies showed that the continuum task, such as the visual analogue scaling (VAS) task is a better measure. See for example:

- Kapnoula, E. C., Winn, M. B., Kong, E. J., Edwards, J. & McMurray, B. (2017). Evaluating the Sources and Functions of Gradiency in Phoneme Categorization: An Individual Differences Approach. Journal of Experimental Psychology: Human Perception and Performance, 43(9), 1594–1611. https://doi.org/10.1037/xhp0000410

Procedure: The authors may need to clarify whether there is any break between each phase.

Line 256: Please clarify why a sample size of 50 can ensure sufficient power for EFA.

Validity of the findings

Throughout the paper, it may be better to focus on the main experiment but only briefly talk about the pilot. Given that the sample size is small in the pilot, it is difficult to judge how robust the results are. Also, doing that can leave more space to talk about the theoretical contribution and other details of the main experiment and the current study.

Line 293: Please clarify which variables acted as the random slope of participants. Only the fixed factor test phase or also the interaction?

Line 366-368: I don’t understand where the effect size came from. It seems that the interaction effect was not tested in pilot?

Line 416-417: Did you imply that an objective language proficiency measure may be better than a subjective one? Also, in the pre-registration, you mentioned that besides self-reported Mandarin proficiency, another measure, accuracy at identifying fricatives and affricates that differ in retroflexion in the Beijing Mandarin variety, would be collected. But it seems that this did not appear in the manuscript. Any reason for that?

---

## Round 0.2 · accepted · Accept

Reviewer 1 has noted a few minor issues that you can fix at the proofs stage.

Reviewer 1 ·

Basic reporting

Minor typos:
The first sentence of the Participants section for the pilot study has "thestudent"-->"the student"

P 23, line 405 of pdf: "However,," --> "However,"

A link to the osf page with pre-registration (osf.io/s6vdn) within text at least once would be easier for the reader than just finding it in the references. Also note the reference states that the pre-registration is embargoed until 2024, but since it is public now, this can be left off.

On a similar note, one direct link to the repository page (which is a different osf link than the pre-registration: osf.io/agrpj) in text would also make things easier for the reader rather than having to find it through the pre-registration.

Experimental design

no comment

Validity of the findings

no comment

Additional comments

The authors have addressed all my comments satisfactorily. Thank you for this interesting paper!

·

Basic reporting

no comment

Experimental design

no comment

Validity of the findings

no comment

Additional comments

The authors have done a great job on addressing all of my previous comments and I have nothing to add.